# Multi-omics evaluation of cell lines as models for metastatic prostate cancer
Xueying Liu[1], Weixing Yu[2], Xiuyuan Jin[1], Yugang Wang [2]✉ & Ke Liu [1]✉

Cell lines are indispensable tools in prostate cancer research yet their suitability as models for distance metastasis remains incompletely characterized. Here, we conduct a systematic evaluation study using large-scale public multi-omics data. We reveal substantial genomic differences between cell lines and metastatic patient samples, and meanwhile pinpoint cell lines which more closely resemble metastatic prostate cancer. Notably, hypermutation significantly influences the tumor microenvironment, underscoring the importance of considering mutational burden in model selection. Surprisingly, the widely used PC3 cell line exhibits poor transcriptomic and epigenomic similarity to any prostate cancer subtype, revealing a previously unrecognized limitation. Furthermore, we find existing engineered stem-like cell lines fail to faithfully recapitulate the transcriptomic profiles of mesenchymal stem-like prostate cancer, whereas selected organoids exhibit higher fidelity. Our study provides guidance for cell line selection in metastatic prostate cancer research and highlights the urgent need to develop improved cell lines for the mesenchymal stem-like subtype.

Cancer cell lines are indispensable preclinical models in oncology research. Their immortality, cost-effectiveness, and capacity for long-term cultivation and cloning make them vital for reproducible and scalable cancer research and drug development[1–5]. However, prolonged in vitro culture can lead to the accumulation of additional genetic and epigenetic alterations, resulting in substantial discrepancies compared to patient-derived tumor samples[6–9]. Therefore, systematic evaluation of the molecular fidelity of cancer cell lines relative to patient-derived cancer cells is essential to ensure the translational relevance of cell line–based studies.

The widespread availability of large-scale open cancer genomics data has enabled researchers to comprehensively assess the suitability of cell lines as models of cancer research. For example, Domcke et al.[10] compared genomic profiles between ovarian cancer cell lines and patient tumor samples and found several commonly used cell lines poorly recapitulated high-grade serous ovarian cancer; Yu et al.[11] performed a pan-cancer transcriptomic comparison between cell lines and patient tumor samples and proposed a new cell line panel TCGA-110-CL as a replacement for the canonical NCI-60 panel; Jin et al.[12] conducted similar pan-cancer analysis with their own computational approaches to select appropriate cell line models for different cancer types; We previously performed multi-omics comparison between breast cancer cell lines and metastatic tumor samples and identified cell lines that faithfully represent distinct breast cancer subtypes[13]. These studies have deepened our

understanding of cancer cell lines and highlighted the importance of selecting appropriate models based on molecular fidelity.

Prostate cancer is among the most common and lethal malignancies affecting men's health worldwide, posing a substantial public health challenge[14,15]. Similar to other cancer types, cell lines have also been widely utilized to explore the underlying biological mechanisms of this disease. Although several studies have evaluated the molecular representativeness of prostate cancer cell lines relative to tumor samples[11,12,16], three major limitations persist. First, metastasis is the leading cause of prostate cancer–related mortality[14,17]; however, most of the evaluation studies mainly focus on primary tumors. Second, previous studies were performed in a pan-cancer manner, which applies uniform analytical criteria across various cancer types and overlooks the influence of prostate cancer-specific characteristics in cell line selection[3]. Third, the majority of analyses rely exclusively on transcriptomic data, with limited integration of multi-omics information for a more comprehensive evaluation[18].

In this study, we leveraged the publicly available bulk and single-cell omics data to comprehensively evaluate the suitability of prostate cancer cell lines as models for metastatic prostate cancer. We proposed principles of cell line selection for different research scenarios, and meanwhile, point out the strengths and limitations of widely used cell lines. Our study provides a framework for selecting prostate cancer cell lines tailored to specific biological contexts and highlights the need to develop more representative models for understudied prostate cancer subtypes.

[1]Department of Medical Dataology, School of Public Health, Cheeloo College of Medicine, Shandong University, Jinan, China. [2]Department of Biochemistry and Molecular Biology, School of Basic Medicine, Tongji Medical College, Huazhong University of Science and Technology, Wuhan, China. ✉e-mail: yugangw@hust.edu.cn; keliu.iluke@email.sdu.edu.cn

## Results

### Systematic comparison of genomic profiles

We first compared the somatic mutation profiles between metastatic prostate cancer samples and prostate cancer cell lines, focusing on genes that were either highly mutated in metastatic prostate cancer or exhibited differential mutation rates between metastatic and primary prostate cancer. The somatic mutation data of primary prostate cancer, metastatic prostate cancer, and prostate cancer cell lines were downloaded from the Cancer Genome Atlas (TCGA), Stand Up to Cancer (SU2C), and the Cancer Cell Line Encyclopedia (CCLE), respectively[19–21]. A total of 68 highly mutated genes were identified using SU2C samples, along with 13 differentially mutated genes between SU2C and TCGA samples (Supplementary Fig. 1a–c). The integration of these two lists identified 73 unique genes (Fig. 1a). The median mutation rate of the 73 genes across the CCLE prostate cancer cell lines was 0.23, suggesting most of them could be recapitulated by only a few cell lines. *MUC16*, *SYNE2*, *SYNE1*, and *OBSCN* have a mutation rate higher than 50%, with *MUC16* being the most frequently mutated gene (mutation rate = 0.62). *TP53*, a key tumor suppressor gene, displayed a mutation rate of 0.38. Surprisingly, there were four genes (*APC*, *SPOP*, *CDK12*, and *FAM186A*) that were entirely unmutated in all analyzed prostate cell lines, highlighting the limitation of using the analyzed cell lines in researching the role of these genes in metastasis-related studies.

Cell lines were ranked based on the number of mutations present in a set of 73 genes associated with metastatic prostate cancer, and found the top four cell lines were LNCaP, DU145, 22RV1, and MDA-PCa-2b, respectively (Fig. 1b). Interestingly, previous studies have inferred that these four cell lines exhibit microsatellite instability (MSI)[22], which is consistent with their markedly elevated mutation burden compared with other prostate cancer cell lines (Fig. 1c). To adjust such bias, we re-ranked the cell lines according to a normalized ratio value (computed as number of mutated genes/mutation burden) and found the top three cell lines were PC3, VCaP, and NCI-H660 (Fig. 1d).

We next investigated whether those recurrent single-nucleotide alterations (which we call "hotspot mutation") identified in metastatic prostate cancer samples were also present in prostate cancer cell lines. We defined hotspot mutations as non-synonymous mutations occurring in at least three SU2C samples. Based on this criterion, 58 hotspot mutations were identified (Supplementary Data 1). These hotspot mutations were distributed in 45 genes, with *TP53* and *AR* being the two genes harboring the most hotspot mutations (Fig. 1e and Supplementary Fig. 1d). *TP53* hotspot mutations were predominantly located within the DNA-binding domain, a functionally critical region required for the transcriptional regulation of tumor suppressor genes[23]. For *AR*, hotspot mutations were primarily clustered within the ligand-binding domain, which is pivotal for androgen receptor-targeted therapies[24].

Strikingly, only seven hotspot mutations were detected in at least one prostate cancer cell line (Fig. 1f), and all were presented exclusively in 22RV1, MDA-PCa-2b, and LNCaP. Among them, three were single-nucleotide point mutations of *AR*, and the other four were all frameshift mutations. Of the *AR* hotspot mutations, T878A and L702H mutations have been reported to be associated with resistance to AR inhibition therapy[25]. Notably, MDA-PCa-2b harbors both mutations (T878A and L702H), reflecting the clinical phenomenon of co-occurring multiple *AR* resistance mutations within individual tumors[25]. Among the frameshift hotspot mutations, the K16fs mutation in *RPL22* was observed in all three cell lines, whereas the other three frameshift mutations were exclusive to 22RV1.

Besides somatic mutation profiles, we also compared copy number variation (CNV) profiles between metastatic and primary prostate cancer samples. Among the differential CNV genes between SU2C and TCGA samples, *AR* exhibited the highest level of amplification (Fig. 1g). A similar pattern was observed in site-specific analyses of differential CNV genes (Supplementary Fig. 1e–g). This finding may not be surprising since *AR* is a critical driver of prostate cancer development and progression[26–28]. Notably, VCaP exhibited the highest level of *AR* amplification among all analyzed prostate cancer cell lines (Fig. 1h).

### Hypermutated prostate cancer has increased cytotoxic CD8+ T cell infiltration

Previous studies have demonstrated hypermutated colorectal tumor samples exhibiting increased levels of tumor-infiltrating lymphocytes, elevated neoantigen burden, and potentially enhanced sensitivity to immunotherapy[29,30]. Motivated by these findings, we aim to assess whether hypermutation should be taken into consideration when selecting cell lines for studying the metastatic mechanisms of prostate cancer.

To identify hypermutated prostate cancer samples, we aggregated somatic mutation data from TCGA, SU2C, and CCLE and ranked all samples by mutation burden. A pronounced gap in mutation burden was observed between the 798th and 799th ranked samples, indicating a natural cutoff for selection of hypermutated samples (Fig. 2a). This threshold was further supported by the bimodal distribution of mutation burden, reinforcing the robustness of the classification (Supplementary Fig. 2a). Applying this threshold, we identified four hypermutated samples in TCGA, ten in SU2C, and four in CCLE. As expected, the aforementioned four hypermutated cell lines, LNCaP, DU145, 22RV1, and MDA-PCa-2b, were successfully identified. Notably, three of them harbored *RPL22* frameshift mutations, which are frequently observed in MSI tumors[31,32]. To further validate our strategy, we computed the mutation frequencies of mismatch repair (MMR) genes[33] and found hypermutated samples exhibiting a significantly higher mutation frequency (Fig. 2b and Supplementary Fig. 2b). These results demonstrate the validity of the determined threshold value for identifying hypermutated prostate cancer samples.

We next performed differential gene expression analysis between hypermutated and non-hypermutated TCGA samples and identified 125 upregulated and 85 downregulated genes (Supplementary Data 2). Notably, genes encoding T cell markers (*CD3D* and *CD3E*), cytotoxic CD8⁺ T cell markers (*CD8A*, *GZMA*, *GZMB*, *PRF1*), and immune checkpoint molecules (*PDCD1* and *CTLA4*) were consistently upregulated in the hypermutated group (Fig. 2c), suggesting increased cytotoxic CD8⁺ T cell infiltration in the tumor microenvironment (TME) of hypermutated prostate cancer. To further validate this observation, we applied the Wilcoxon rank-sum test to these genes and obtained consistent results (Supplementary Fig. 2c). In addition, we employed two independent tools, TIDE[34] and xCell[35], to estimate cytotoxic CD8⁺ T cell infiltration levels, both of which showed higher average infiltration in the hypermutated group, although the statistical significance was marginal—likely due to the limited sample size of hypermutated cases (Supplementary Fig. 2d). Finally, enrichment analysis of the upregulated genes using the MSigDB Hallmark gene sets revealed six significantly enriched pathways, including "Interferon Gamma Response" (Fig. 2d and Supplementary Data 3), further supporting our findings.

The TME has been demonstrated to play pivotal roles in the distant metastasis of primary cancer[36–38], and our analysis suggests notable TME differences between hypermutated and non-hypermutated primary prostate cancer samples. Therefore, we propose that hypermutated cell lines (such as LNCaP) should be more appropriate for investigating the metastatic mechanisms of hypermutated prostate cancer. To the best of our knowledge, this consideration has not been addressed in previous pan-cancer cell line evaluation studies, highlighting the need for cancer-specific assessments.

### Correlating cell lines with metastatic prostate cancer using bulk and single-cell RNA-seq data

Transcriptomic correlation analysis (TC analysis) has been proven to be an effective method for assessing the utility of cell lines[11,13,39]. Consequently, we ranked all 1,019 CCLE cell lines based on their transcriptomic similarity to MET500 prostate cancer samples, and the top three cell lines were LNCaP, VCaP, and MDA-PCa-2b, respectively (Fig. 3a and Supplementary Data 4). We next examined whether the metastatic site influenced TC analysis results and found that cell line rankings were highly consistent among various metastatic sites (bone, liver, and lymph node) (Fig. 3b and Supplementary Fig. 3a–c).

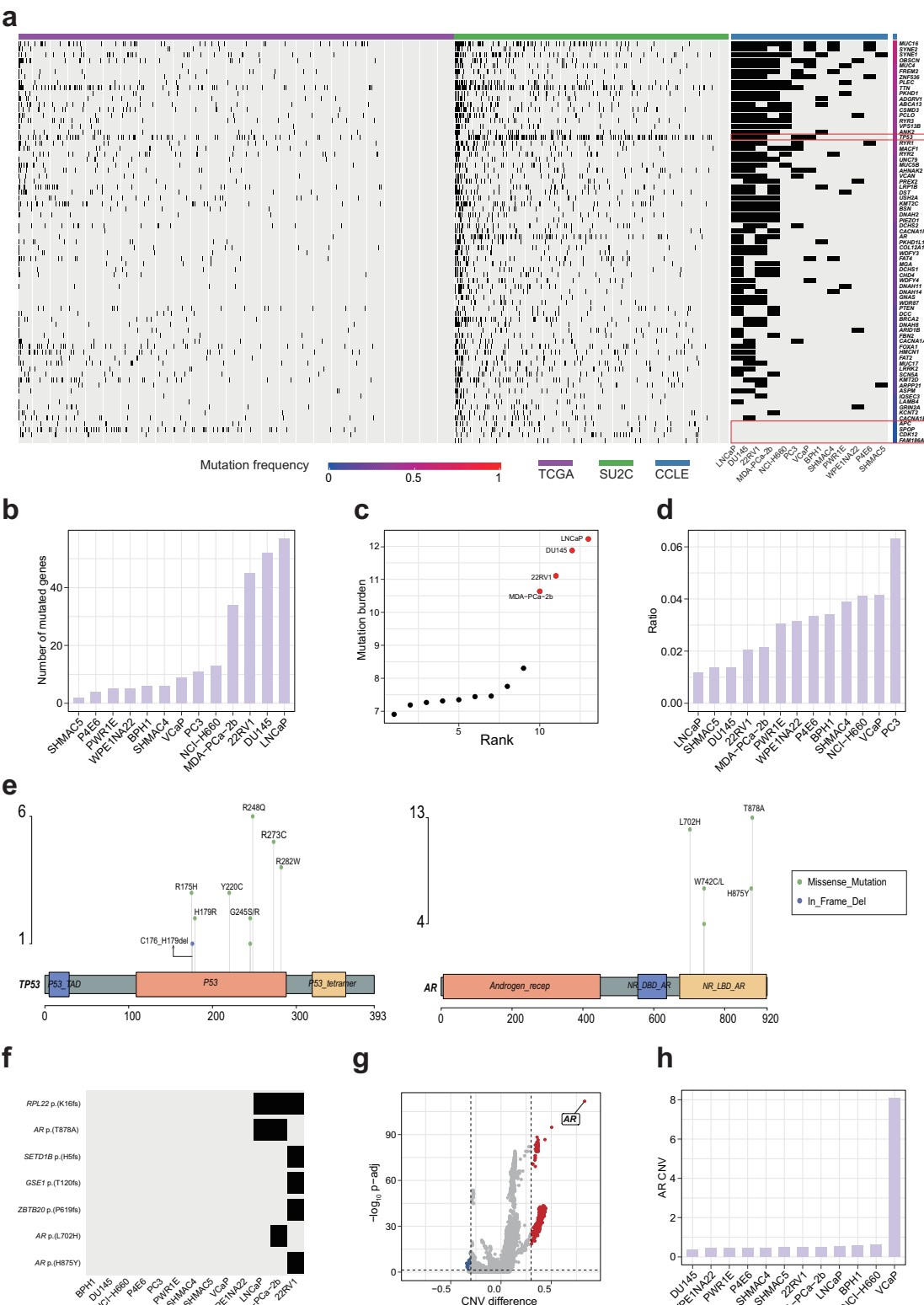

**Fig. 1 | Systematic comparison of genomic profiles. a** Somatic mutation landscape of the 73 selected genes across TCGA (primary), SU2C (metastatic), and CCLE (cell line) samples. The top annotation bar indicates the data source, and the right bar shows the mutation frequency of each gene. *TP53*, *APC*, *SPOP*, *CDK12*, and *FAM186A* are highlighted with red boxes. **b** Ranking CCLE prostate cancer cell lines based on the total number of mutations in the 73 genes. **c** Dot plot of mutation burden in CCLE prostate cancer cell lines. **d** Ranking CCLE prostate cancer cell lines based on normalized ratio (mutation count adjusted by mutation burden). **e** Lollipop plots showing mutation hotspots in *TP53* (left) and *AR* (right). The protein domains are annotated according to PFAM. **f** Presence (or absence) of mutation hotspots across CCLE prostate cancer cell lines. **g** Volcano plot comparing copy number variation (CNV) between SU2C and TCGA samples. Each point represents a gene; the x-axis denotes the difference in median CNV values (SU2C − TCGA), and the y-axis shows statistical significance (−log$_{10}$(adjusted *P*-value)). Dashed vertical lines represent CNV difference thresholds (±0.3), and the horizontal dashed line indicates an adjusted *P*-value cutoff of 0.05. **h** CNV value of the *AR* gene across CCLE prostate cancer cell lines.

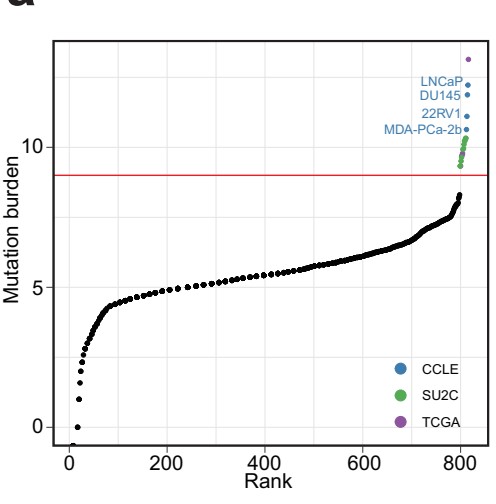

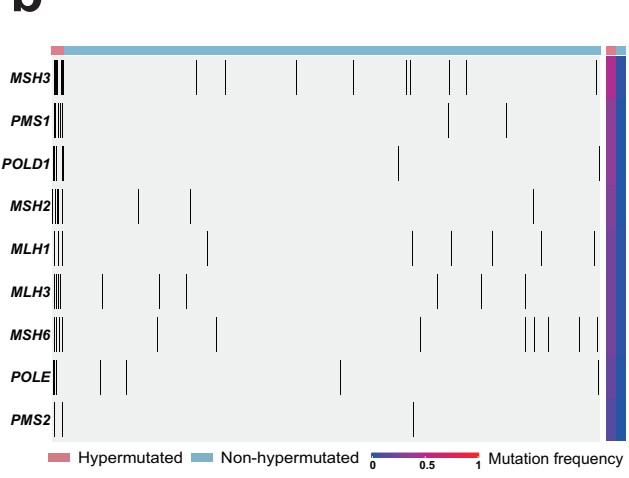

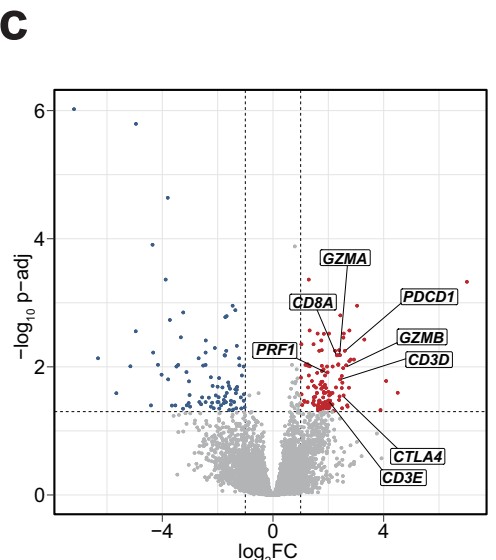

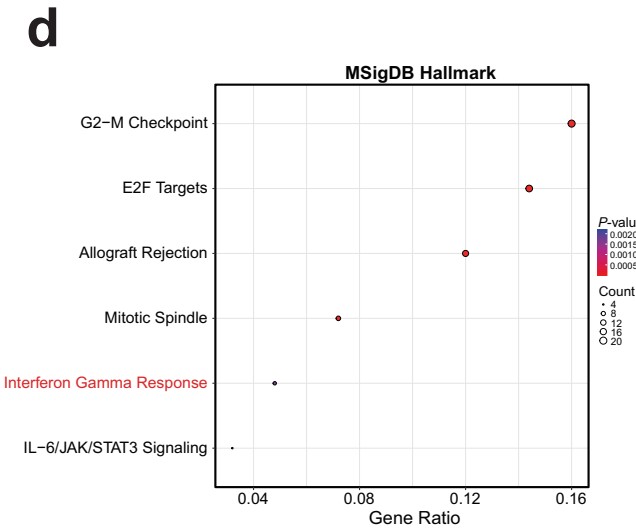

**Fig. 2 | Hypermutated prostate cancer has increased cytotoxic CD8$^+$ T cell infiltration. a** Identification of hypermutated samples. Samples from TCGA, SU2C, and CCLE are combined and ranked by total mutation burden. The red line denotes the threshold used to define hypermutation. Hypermutated samples are color-coded by dataset origin, and hypermutated CCLE cell lines are labeled. **b** Somatic mutation profile of mismatch repair (MMR) genes in hypermutated versus non-hypermutated samples. The top annotation bar indicates sample classification (hypermutated vs. non-hypermutated), while the right annotation bar shows the mutation frequency of each MMR gene. **c** Volcano plot of differentially expressed genes between hypermutated and non-hypermutated TCGA samples. Red and blue dots indicate significantly upregulated and downregulated genes, respectively (adjusted $P$-value < 0.05, |log$_2$FC| > 1). The x-axis shows log$_2$ fold change; the y-axis shows $-$log$_{10}$(adjusted $P$-value). Dashed vertical lines indicate |log$_2$FC| = 1, and the horizontal line marks the significance threshold of $P$-value = 0.05. **d** Enrichment analysis of the 125 genes significantly upregulated in hypermutated TCGA samples using hallmark gene sets from the MSigDB database.

To double confirm the analysis results derived from bulk RNA-seq data, we further ranked 1,019 CCLE cell lines according to their transcriptomic similarity with the malignant cells from a Smart-seq2 scRNA-seq dataset of metastatic castration-resistant prostate cancer (mCRPC)[26]. Consistent with previous results, the top three cell lines with the highest similarity were VCaP, LNCaP, and MDA-PCa-2b (Fig. 3c). We further performed metastatic-site-specific transcriptomic similarity analysis (bone and lymph node), and the results were highly correlated (Fig. 3d and Supplementary Fig. 3d, e).

We conducted a PubMed search of studies related to metastatic prostate cancer and found that the three most frequently cited cell lines (PC3, LNCaP, and DU145) accounted for over 90% of all citations

(Supplementary Fig. 3f and Supplementary Data 5). Surprisingly, VCaP, which demonstrates both high genomic and transcriptomic similarity to metastatic prostate cancer samples in our analysis, was cited in only 1.4% of studies. This discrepancy highlights a notable mismatch between the actual suitability of cell lines and their prevalence in the literature, and underscores the importance of cell line evaluation for improving translational relevance in prostate cancer research.

### Limitations of the PC3 cell line
According to PubMed search, PC3 is the most widely used cell line in studies investigating metastatic prostate cancer[40,41]. However, our analysis suggests that this cell line shows limited similarity to metastatic prostate cancer

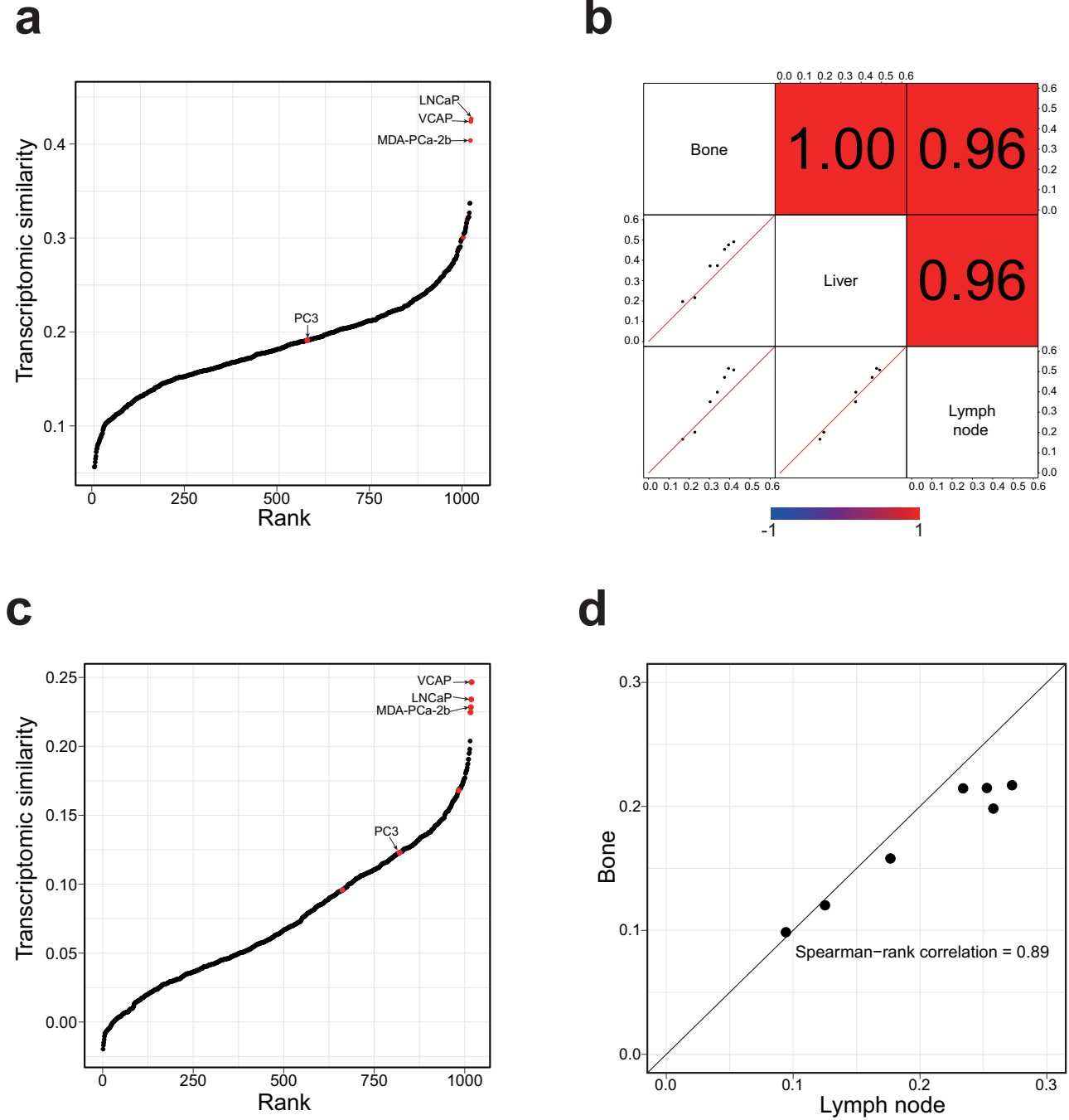

**Fig. 3 | Correlating cell lines with metastatic prostate cancer using bulk and single-cell RNA-seq data. a** Ranking 1019 CCLE cell lines based on their transcriptomic similarity to MET500 prostate cancer samples. Each dot represents a CCLE cell line, and the prostate cancer cell lines are highlighted in red. **b** Pair-wise comparison of site-specific TC analysis results. In the lower-left plots, each dot is a CCLE prostate cancer cell line, with the two axes representing transcriptomic similarity to MET500 prostate cancer samples of the two intersecting sites. The upper-right shaded values are the corresponding pair-wise Spearman rank correlation values of each pair. **c** Ranking 1019 CCLE cell lines based on their transcriptomic similarity to the malignant cells from a metastatic prostate cancer scRNA-seq dataset. Each dot represents a CCLE cell line, and the prostate cancer cell lines are highlighted in red. **d** TC analysis results using single malignant cells from different metastatic sites are highly correlated. Each dot represents a CCLE prostate cancer cell line, with the x-axis and y-axis representing transcriptomic similarity to the malignant cells from lymph node and bone metastases, respectively.

samples (ranked 576 among 1019 CCLE cell lines) (Fig. 3a). Prior to our research, several studies had revealed the heterogeneity of prostate cancer[42–44]. Considering that the metastatic prostate cancer samples used in the transcriptomic correlation analysis mainly exhibited adenocarcinoma histology (in which malignant cells primarily show luminal differentiation)[45–47], we further investigated whether PC3 could serve as a representative model for other subtypes of prostate cancer.

We performed principal component analysis (PCA) on gene expression profiles of CCLE prostate cancer cell lines and found they could be briefly classified into three clusters (Fig. 4a). Cluster 1 consisted of luminal lineage cell lines (such as VCaP[48]), as indicated by expression of luminal lineage markers such as *AR* and *KLK4*; cluster 2 only included NCI-H660[49], a small cell carcinoma cell line of neuroendocrine origin expressing *SYP*; cluster 3 comprised DU145 and PC3. Interestingly, our PCA results align

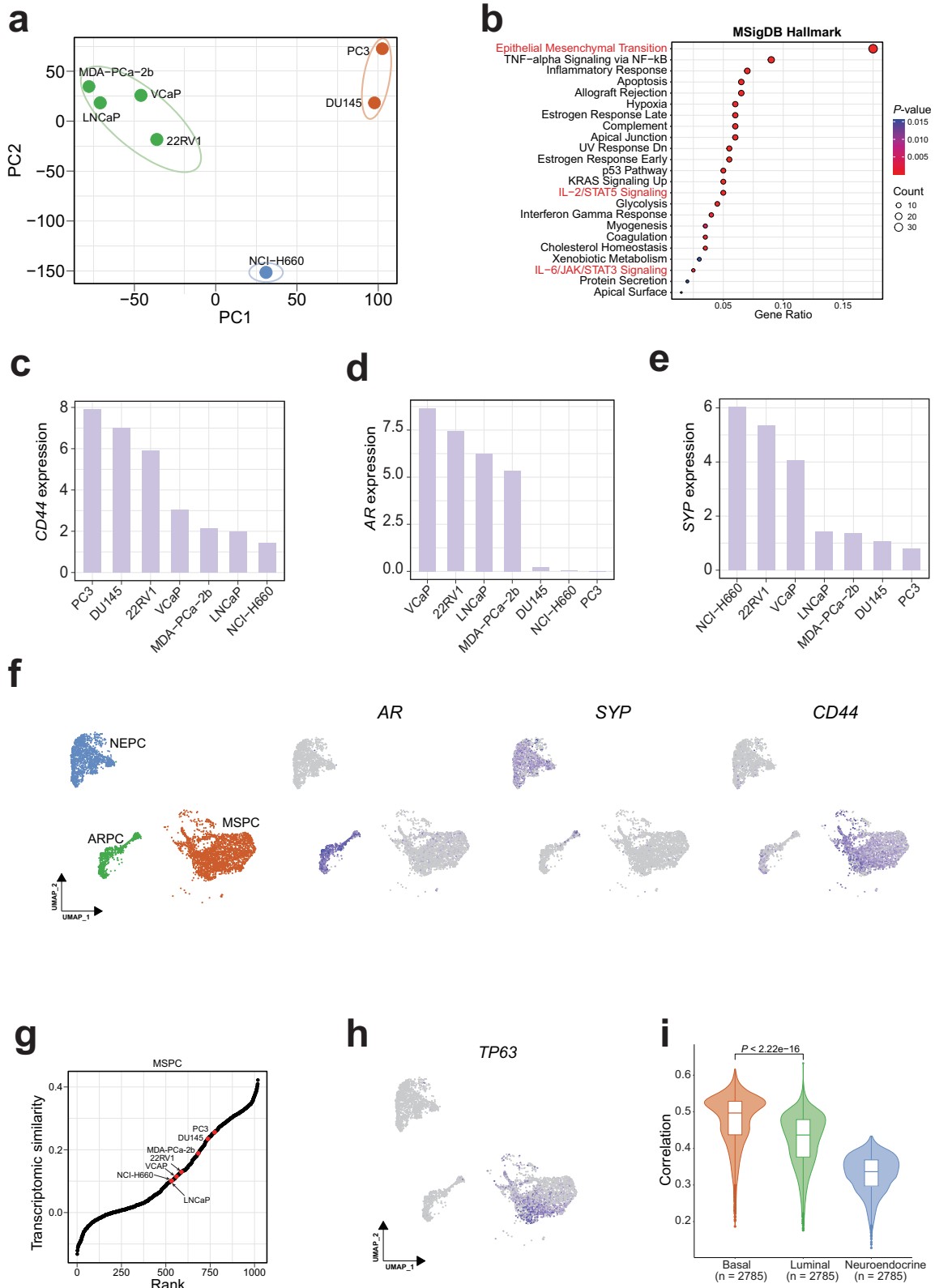

with Han et al.'s research, which classified prostate cancer into three intrinsic subtypes: androgen receptor positive prostate cancer (ARPC), neuroendocrine prostate cancer (NEPC), and mesenchymal and stem-like prostate cancer (MSPC)[50]. ARPC represents the majority of prostate adenocarcinomas and is defined by luminal differentiation and strong androgen receptor signaling. NEPC is characterized by the expression of canonical neuroendocrine markers. Notably, MSPC is characterized by loss of AR signaling and acquisition of mesenchymal (and stem-like) transcriptional features, and is often associated with lineage plasticity and therapeutic resistance[50]. Since PC3 and DU145 were separated from other cell lines on the first principal component (PC1), we performed enrichment analysis on the top 200 genes with the highest positive loadings along the PC1 axis using

**Fig. 4 | Limitations of the PC3 cell line. a** Principal Component Analysis (PCA) of CCLE prostate cancer cell lines based on gene expression profiles. Three distinct clusters are identified and color-coded. **b** Enrichment analysis of the top 200 genes with the highest positive loadings along PC1 using hallmark gene sets from the MSigDB database. Expression of key lineage markers *CD44* (**c**), *AR* (**d**), and *SYP* (**e**) across CCLE prostate cancer cell lines. **f** UMAP visualization of the malignant cells from a castration-resistant prostate cancer (CRPC) scRNA-seq dataset (accession number: PRJNA699369). The left panel shows subtype annotations; the right three panels depict expression levels of *AR*, *SYP*, and *CD44*, respectively, and color represents expression level, from gray (low) to dark blue (high). **g** Ranking 1019 CCLE cell lines based on their transcriptomic similarity to the MSPC malignant cells. Each dot represents a CCLE cell line, and the prostate cancer cell lines are highlighted in red. **h** Expression of *TP63* in malignant cells from the CRPC scRNA-seq dataset. Color represents expression level, from gray (low) to dark blue (high). **i** Transcriptomic correlation between MSPC malignant cells and three normal prostate epithelial cell types. In each box, the central line represents the median value and the bounds represent the 25th and 75th percentiles (interquartile range). The whiskers encompass 1.5 times the interquartile range. Outliers are shown as individual points. *P*-value was calculated using the two-sided Wilcoxon signed-rank test.

MSigDB Hallmark gene sets (Fig. 4b and Supplementary Data 6). The "Epithelial-Mesenchymal Transition" gene set exhibited the highest level of enrichment; in addition, the enrichment of "*IL-2/STAT5* Signaling" and "*IL-6/JAK/STAT3* Signaling" gene sets suggests activation of the *JAK/STAT* pathway, a known driver of cellular plasticity[47,51]. Furthermore, PC3 exhibited high expression of the stemness marker *CD44*, virtually no expression of *AR*, and only low-level expression of the neuroendocrine marker *SYP* (Fig. 4c–e), which can be double confirmed in additional PC3 bulk RNA-seq and scRNA-seq datasets (Supplementary Fig. 4a–e). Collectively, these results support the classification of PC3 as a cell line of the MSPC subtype.

We next evaluated whether PC3 could adequately recapitulate the transcriptomic profile of MSPC malignant cells. MSPC has been reported to be enriched in CRPC patients[50]. To this end, we analyzed a CRPC scRNA-seq dataset and found that the malignant cells could be classified into ARPC, NEPC, and MSPC subtypes based on the expression of *AR*, *SYP*, and *CD44*, respectively (Fig. 4f). We then ranked 1,019 CCLE cell lines according to their transcriptomic similarity with malignant cells in a subtype-specific manner. As expected, MDA-PCa-2b, VCaP, and LNCaP showed the highest similarity to ARPC, while NCI-H660 (a neuroendocrine prostate cancer cell line) exhibited a high similarity to NEPC (Supplementary Fig. 4f, g). Strikingly, for the MSPC subtype, PC3 only got a rank value of 775, although its rank was the highest among prostate cancer cell lines (Fig. 4g and Supplementary Data 7). Moreover, the transcriptomic correlation between individual MSPC malignant cells and PC3 was significantly lower than that between ARPC malignant cells and VCaP, suggesting limited fidelity of PC3 as a model of MSPC (Supplementary Fig. 4h). Interestingly, we found MSPC malignant cells expressed basal lineage markers such as *TP63* and *CAV2* (Fig. 4h and Supplementary Fig. 4i), suggesting their basal differentiation. Consistent with this, lineage analysis suggested their transcriptomic profile was most similar to basal cell (Fig. 4i and Supplementary Fig. 4j). However, PC3 lacked expression of key basal markers (such as *TP63*), which may explain its low transcriptomic similarity with MSPC malignant cells (Supplementary Fig. 4k, l).

To rule out the possibility that the unexpectedly low ranking of PC3 was driven by batch effects in the CCLE dataset, we merged the previously mentioned additional PC3 samples with CCLE samples and re-performed the analysis (Supplementary Fig. 4m). Although the mean rank of these external PC3 samples was higher than that of the CCLE PC3 sample, their overall ranking remained modest relative to the 1019 CCLE cell lines. In addition, we also utilized another scRNA-seq dataset of mCRPC[47] (which contains the malignant cells of all three subtypes) to re-perform the analysis and derived similar results (Supplementary Fig. 5a–d).

To further validate our findings, we also performed the correlation analysis based on chromatin-accessibility profiles. We searched the public domain and collected ATAC-seq data for six prostate cancer cell lines and seventy metastatic prostate cancer patient samples[52]. We first applied PCA on prostate cancer cell lines and found three distinct clusters corresponding to ARPC (VCaP, LNCaP, 22RV1), NEPC (NCI-H660), and MSPC (PC3, DU145) subtypes, which closely aligned with the PCA results derived from RNA-seq data (Fig. 5a). We next performed functional enrichment analysis using GREAT[53] on the 1000 peaks with the highest positive loadings along the PC1 axis. As expected, "Epithelial-Mesenchymal Transition" and *JAK/*

*STAT* signaling related pathways (such as "*IL-2/STAT5* Signaling" and "*IL-6/JAK/STAT3* Signaling") were significantly enriched, mirroring our previous enrichment analysis results (Fig. 5b). We next subtyped patient samples based on their matched bulk RNA-seq profiles (Supplementary Fig. 5e–g) and then quantified their chromatin-accessibility similarity with prostate cancer cell lines. As expected, the results were highly concordant with those derived from transcriptomic correlation analysis: VCaP showed the strongest similarity to ARPC samples, while NCI-H660 most closely aligned with NEPC samples. However, PC3 exhibited moderate similarity to MSPC samples and was even significantly lower than NCI-H660 (Fig. 5c–e).

Taken together, our findings highlight the limitations of the PC3 cell line. Although it displays molecular features of the MSPC subtype—such as elevated EMT and stemness programs—its overall transcriptomic and chromatin-accessibility similarity to MSPC malignant cells remains modest.

## Patient-derived organoids better resemble the transcriptome of MSPC than PC3

Since PC3 does not fully recapitulate the transcriptomic features of MSPC, we next investigated other models that could be more representative. Previous studies have reported that AR inhibition by enzalutamide or *TP53/RB1* double-knockout can promote LNCaP (a hypermutated cell line of ARPC) into the MSPC state[50,51]. Therefore, we included these two engineered LNCaP-derived models in our analysis. In addition, we generated a VCaP-derived cell line by overexpressing *MET* (VCaP-MET-OE), as *MET* has been reported to promote EMT and sustain stemness in cancer cells[54–56]. As expected, bulk RNA-seq analysis suggested that *MET* overexpression induced upregulation of *CD44* (log$_2$FC = 0.251, *P*-value = 0.028) and increased *JAK/STAT* pathway activity, while downregulating luminal markers *KLK2*, *KLK3*, and *KLK4*, suggesting the occurrence of dedifferentiation (Fig. 6a, b and Supplementary Data 8).

We next computed the transcriptomic similarity with MSPC malignant cells for the above three engineered cell lines, respectively. Surprisingly, all of the engineered cells had significantly lower transcriptomic similarity with MSPC malignant cells than PC3 (Fig. 6c). In addition, lineage analysis revealed that these engineered cell lines retained their original luminal differentiation (Fig. 6d). Furthermore, when comparing the transcriptomic profiles of these engineered cell lines to the 1,019 CCLE cell lines, the highest correlations were consistently observed with their respective parental lines (Supplementary Fig. 6a–g), suggesting the engineering did not trigger a luminal-basal lineage transition. MSPC malignant cells exhibit basal differentiation, and this likely explains why the engineered cell lines failed to recapitulate the MSPC transcriptional landscape.

In recent years, patient-derived organoids (PDOs) have gained increasing attention for studying cancer biology. We next explored whether this type of model could better recapitulate the MSPC transcriptome. We computed the transcriptomic similarity between ten established PDOs (all of which were claimed as MSPC according to Tang et al.'s study)[57] and MSPC malignant cells. Notably, four PDOs had significantly higher transcriptomic similarity to MSPC malignant cells than PC3, with MSKPCa12 being the highest one (Fig. 6e). To further validate this finding, we also computed the chromatin-accessibility similarity between the ten PDOs and MSPC samples. Consistent with the transcriptomic analysis results,

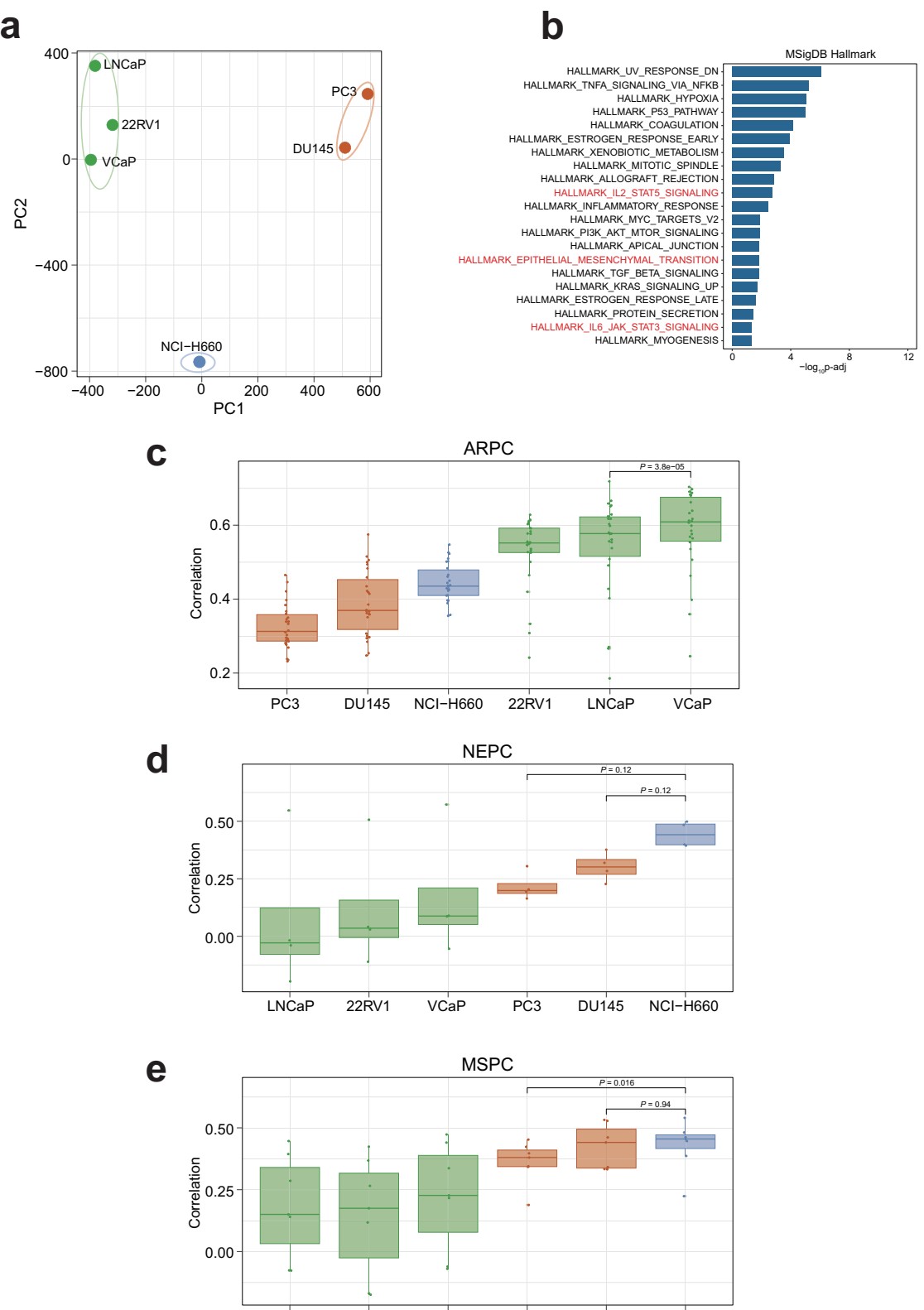

**Fig. 5 | Correlation analysis based on ATAC-seq data. a** PCA of prostate cancer cell lines based on ATAC-seq data. Three distinct clusters are identified and color-coded. **b** Enrichment analysis of the top 1000 peaks with the highest positive loadings along PC1 using GREAT. Only significant terms (adjusted *P*-value < 0.05) are shown. **c–e** Ranking prostate cancer cell lines based on their chromatin-accessibility similarity to patient samples of different subtypes. The sample sizes were 26 for ARPC, 4 for NEPC, and 7 for MSPC. Boxes show the median (central line) and interquartile range (25th–75th percentiles); whiskers extend to 1.5× IQR. Outliers are displayed as individual points. *P*-values were calculated using the two-sided Wilcoxon signed-rank test.

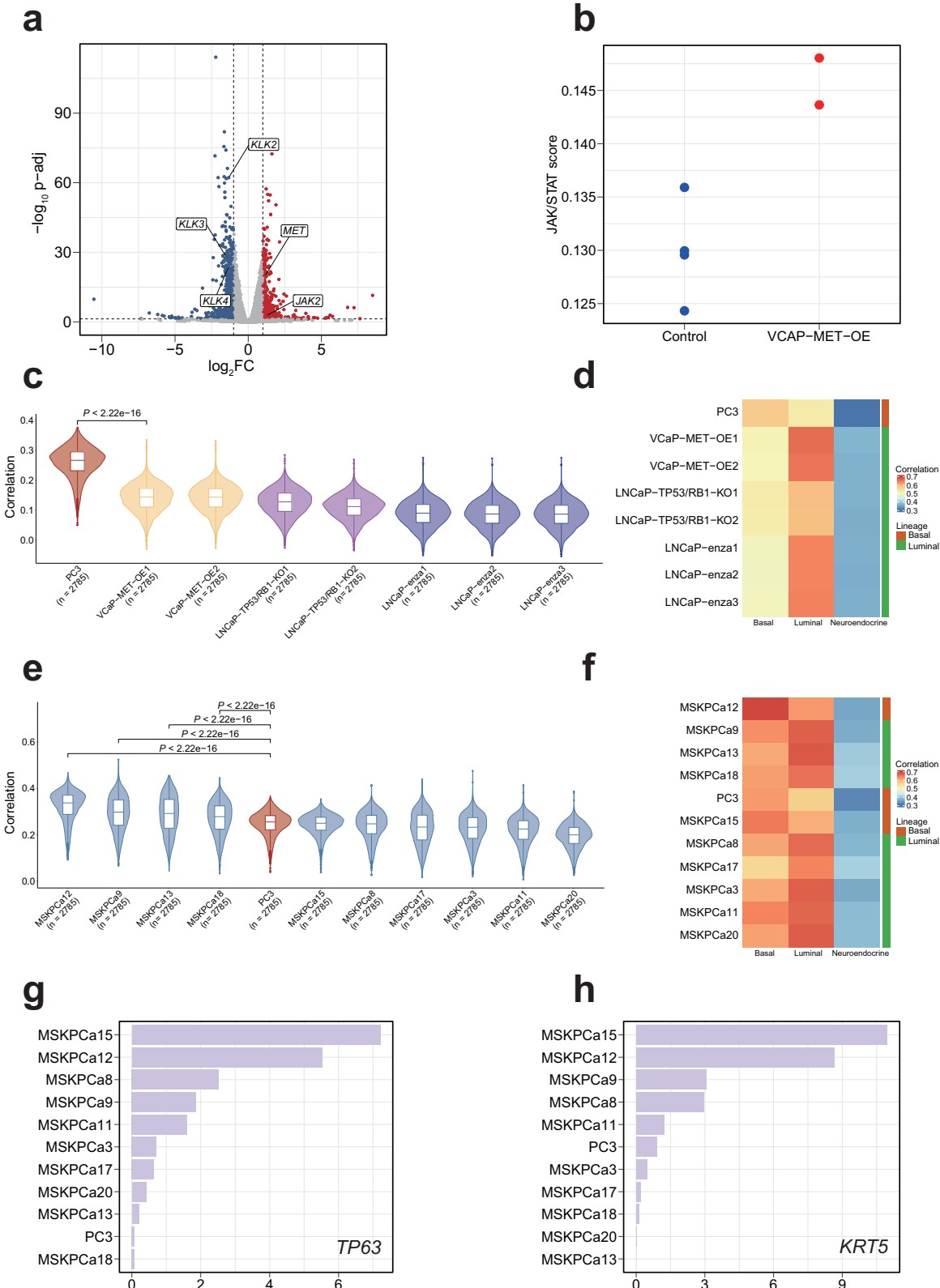

MSKPCa12 displayed significantly higher similarity than PC3 (Supplementary Fig. 6h). Further lineage analysis suggested that this organoid exhibited a basal differentiation phenotype (Fig. 6f). As expected, it displayed higher expression of basal-specific markers (such as *TP63* and *KRT5*) than PC3 (Fig. 6g-h), which may explain its stronger transcriptomic resemblance of MSPC malignant cells.

## Discussion

Cell lines are widely utilized in cancer research, particularly as models for metastatic cancers. Prior studies have demonstrated that cell lines derived from the same tumor type exhibit considerable molecular heterogeneity, with certain cell lines more accurately recapitulating the characteristics of metastatic tumors than others[10,39,58,59]. This underscores the critical

**Fig. 6 | Patient-derived organoids better resemble the transcriptome of MSPC than PC3. a** Volcano plot showing differentially expressed genes between VCaP-*MET*-OE and control groups. Red and blue dots indicate significantly upregulated and downregulated genes, respectively, in the VCaP-*MET*-OE group. **b** Dot plot displaying ssGSEA scores for the *JAK/STAT* signaling pathway in VCaP-*MET*-OE and control groups. **c** Violin plots showing the Spearman correlation between MSPC malignant cells and engineered prostate cancer cell lines. In each box, the central line represents the median value and the bounds represent the 25th and 75th percentiles (interquartile range). The whiskers encompass 1.5 times the interquartile range. Outliers are shown as individual points. The *P*-value was calculated using the two-sided Wilcoxon signed-rank test. **d** Heatmap showing transcriptomic correlation between engineered prostate cancer cell lines and three normal prostate epithelial cell types. Lineage identity was assigned based on the highest correlation. **e** Violin plots showing the Spearman correlation between MSPC malignant cells and patient-derived organoids. Plot elements and statistical analysis as in (**c**). **f** Heatmap showing transcriptomic correlation between patient-derived organoids and three normal prostate epithelial cell types. Lineage identity was assigned based on the highest correlation. Expression of *TP63* (**g**) and *KRT5* (**h**) in patient-derived organoids and PC3 cell line.

importance of selecting cell line models that most faithfully represent the biology of metastatic cancers. Our study proposed useful guidance for cell line selection in the study of metastatic prostate cancer.

Adenocarcinoma is the predominant histological subtype of prostate cancer[60,61]. Based on our multi-omics analysis results, VCaP appears to be the most suitable model for studying metastatic prostate cancer according to its relatively higher concordance with metastatic prostate adenocarcinoma patient samples in multiple aspects (for example, high transcriptomic and chromatin-accessibility similarity with ARPC patient samples, and the highest level of *AR* amplification). However, according to PubMed search results, VCaP is rarely used in metastasis-related research, suggesting a gap between model suitability and actual utilization in the field.

Although VCaP serves as an appropriate model for metastatic prostate adenocarcinoma—which is predominantly characterized by a luminal-differentiated, ARPC transcriptional state—it does not fully recapitulate the biological heterogeneity observed in patients. One notable exception is the hypermutated prostate cancer. Although hypermutation is rare in prostate cancer, comparison between hypermutated and non-hypermutated patient samples revealed remarkable biological differences between them. Our analysis suggests that the microenvironment in hypermutated prostate cancer samples differs markedly from that in non-hypermutated tumors, particularly with respect to increased immune infiltration. This indicates that the hypermutation phenomenon should be considered when selecting cell line models. Given its high mutation burden and strong transcriptomic similarity to metastatic prostate adenocarcinoma samples, LNCaP was identified as a promising model for studying hypermutated prostate cancer. However, it is important to note that, like any cell line model in isolation, LNCaP cannot effectively capture immune or stromal interactions. Its utility lies primarily in modeling tumor-intrinsic features such as high mutational burden (which is driven by genomic instability). To investigate how hypermutation shapes the tumor microenvironment, future studies should employ complementary systems—such as co-culture with immune or stromal cells, or in vivo models—in which LNCaP (or similar hypermutated lines) are examined within a more physiologically relevant cellular context.

Another situation is to study AR resistance. We found that the prostate cancer cell lines LNCaP and MDA-PCa-2b harbor *AR* hotspot mutations associated with clinical resistance, showing a certain degree of concordance with patient observations. Specifically, MDA-PCa-2b carries both the T878A and L702H mutations, which are linked to resistance to flutamide and glucocorticoids, respectively. The T878A mutation was also detected in LNCaP, while 22RV1 harbors another recurrent *AR* hotspot mutation, H875Y. Notably, despite harboring these resistance-associated mutations, all three cell lines remain AR-dependent, suggesting that they are valuable in vitro models for studying AR-driven resistance mechanisms and evaluating novel antiandrogen therapies.

According to our PubMed search results, PC3 is the most widely used prostate cancer cell line; however, it exhibits low transcriptomic similarity to metastatic prostate adenocarcinoma. Although further analysis classified PC3 within the MSPC subtype, it does not faithfully model MSPC at the transcriptomic level, either. According to our lineage analysis, MSPC malignant cells exhibited strong basal differentiation, as indicated by the expression of basal lineage markers (such as *TP63*). Although PC3 also

exhibits basal-like differentiation, it lacks expression of several canonical basal lineage markers. Therefore, we conclude that PC3 is in a "semi-basal differentiation" state, and this may explain its transcriptomic discrepancy with MSPC malignant cells.

It is worth noting that although cell lines remain valuable preclinical models for studying metastatic prostate cancer, their transcriptomic divergence from patient tumors should not be overlooked. We also identified differentially expressed genes between cell lines and patient samples based on scRNA-seq data and performed Hallmark pathway enrichment analysis (Supplementary Fig. 7a–f). Cell lines consistently exhibited upregulation of cell cycle programs (e.g., "Mitotic Spindle," "G2–M Checkpoint," "E2F Targets"), accompanied by downregulation of immune-related (e.g., "TNF-α Signaling via NF-κB," "Interferon Gamma Response") and metabolism-related (e.g., "Hypoxia," "Oxidative Phosphorylation") pathways. These transcriptomic differences likely reflect the absence of microenvironmental pressures in the culture and should be carefully considered when interpreting cell line–based studies, particularly in the context of in vitro experiments.

Besides CCLE cell lines, we also examined the suitability of three engineered cell lines (LNCaP-enza, LNCaP-TP53/RB1-KO, VCaP-MET-OE) and PDOs as models of MSPC. We found that the engineering applied to cell lines could only induce stem-like features while not triggering a full luminal-basal lineage transition. Not surprisingly, these engineered cell lines did not exhibit higher performance in resembling the transcriptome of MSPC than PC3. Notably, although several established prostate cancer organoids (such as MSKPCa12) appear to be better models of MSPC than PC3, cell lines are superior in their cost-effectiveness. It is also important to emphasize that our assessment included only a limited number of engineered cell lines and PDOs, representing a small subset of possible model-generation strategies. Comprehensive validation of MSPC models will require evaluating additional PDOs, PDXs, and other relevant experimental systems.

In summary, our comprehensive analysis identified suitable prostate cancer cell lines for different scenarios of metastatic prostate cancer research (Supplementary Fig. 8) and highlighted the limitations of canonical cell lines. We anticipate that the adoption of our recommended cell lines will enhance the translational relevance of in vitro studies of metastatic prostate cancer and accelerate the development of clinically meaningful therapeutic strategies.

## Methods
### Selection of genes used for genomic profile comparison
The somatic mutation data of metastatic and primary prostate cancer were from SU2C and TCGA, respectively. In our analysis, non-adenocarcinoma samples were excluded. Given a gene, we performed Fisher's exact test to evaluate whether its mutation frequency is significantly different between primary and metastatic prostate cancer samples. Since SU2C contains metastatic samples from three sites (liver, bone, and lymph node), the analysis was performed in a site-specific manner. Only genes that were significant (adjusted *P*-value < 0.05) in at least two sites were retained as differentially mutated genes.

Genes whose mutation frequency exceeded 5% in at least two metastatic sites (in the SU2C cohort) were defined as highly mutated.

## Identification of hotspot mutation

Somatic mutation data from the SU2C cohort were first filtered to remove silent variants, retaining only non-synonymous mutations for downstream analyses. For each mutation, we quantified the number of SU2C samples carrying that variant and defined hotspot mutations as those present in at least three samples.

## Comparison of CNV profiles

We downloaded CNV segment data and then utilized CNTools[62] to generate gene-level copy number. For each gene, the Wilcoxon rank-sum test was applied to compare its CNV profile between SU2C and TCGA samples.

## Computation of cytotoxic CD8+ T cell infiltration

TIDE estimates cytotoxic CD8$^+$ T cell infiltration by calculating the mean expression of five marker genes (*CD8A, CD8B, GZMA, GZMB*, and *PRF1*). Given the simplicity of this algorithm, we implemented it directly. xCell is a gene-signature–based method that uses single-sample gene set enrichment analysis (ssGSEA) to infer the abundance of diverse immune cell types. We downloaded the R implementation from its GitHub repository (https://github.com/dviraran/xCell) and used it for our analyses.

## Transcriptomic correlation analysis with bulk RNA-seq and scRNA-seq data

We used transcriptomic correlation analysis to quantify whole-transcriptome similarity. In our previous research, we identified 1000 genes that are highly varied across CCLE cell lines[13]. Given a cell line and a patient sample (or single cell), we defined their transcriptomic similarity as the Spearman rank correlation across the 1000 genes. The transcriptomic similarity between a cell line and several patient samples (or single cells) was computed as the median correlation value.

## Analysis of bulk ATAC-seq data

Patient samples were first assigned to subtypes based on their bulk RNA-seq profiles. According to the annotations reported in the literature[52], samples labeled $AR + NE-$ were classified as ARPC, whereas those labeled $AR - NE+$ were classified as NEPC. To define the MSPC subtype, we calculated ssGSEA scores for EMT and stemness signatures (Supplementary Data 9)[63] and samples with both scores in the top 25% quartile were designated as MSPC.

Chromatin-accessibility similarity between patient samples and prostate cancer cell lines (and organoids) was assessed using bulk ATAC-seq data. We used the pre-processed merged read-count matrix from a reference dataset[52], which provides normalized ATAC-seq read counts for all samples. The top 1% most variable peaks across samples were selected for downstream analysis. Spearman correlation coefficients were then calculated between each patient sample and each cell line (or organoid) using the normalized read count of these variable peaks, providing a quantitative measure of chromatin-accessibility similarity.

## Differential gene expression analysis

DESeq2[64] and Seurat[65] were used to identify differentially expressed genes for bulk RNA-seq and scRNA-seq data (adjusted *P*-value < 0.05 and abs(log$_2$FC) > 1), and the enrichment analysis was conducted using Enrichr[66]. Terms with a *P*-value < 0.05 were considered statistically significant.

## PubMed search

Our PubMed search was performed on March 11th, 2025. For each prostate cancer cell line, we searched the database using the keyword "[cell line name] + metastasis" and counted the number of returned citations.

## Analysis of scRNA-seq data

In the analysis of the prostate cancer scRNA-seq dataset, we used inferCNV[67,68] to infer copy number variation of epithelial cells (immune cells as reference). CNV values were extracted and subjected to k-means clustering to identify malignant cells. The marker genes used in our analysis were: *PTPRC* (immune cells) and *EPCAM* (epithelial cells).

For subtype assignment of malignant cells, we classified cells according to the expression of established subtype-specific markers: *AR* for ARPC, *SYP* for NEPC, and *CD44* for MSPC.

## Lineage analysis

We analyzed a normal prostate scRNA-seq dataset and identified three epithelial cell types: luminal cell (*KLK2, KLK4, ACPP*), basal cell (*KRT5, TP63, KRT17, KRT15*), and neuroendocrine cell (*ASCL1, FOXA2, MYCN, POU3F2, SIAH2, NCAM1, CHGA, CHGB, SYP, ENO2*)[63,69]. We generated a pseudo-bulk profile for each epithelial cell type and then identified the top 5000 genes that are highly variable across the three cell types. Given a single malignant cell, we used these 5000 genes to calculate its transcriptomic similarity with each of the three cell types and then assigned it to the lineage with the highest similarity value.

## Overexpression of MET in the VCaP cell line

VCaP cells and HEK293T cells were obtained from Pricella Biotechnology Co., Ltd and American Type Culture Collection. VCaP and HEK293T cells were maintained in DMEM (catalog no. SH30285.01, Cytiva Life Sciences). All media were supplemented with 10% FBS (catalog no. 164210-50, Pricella Biotechnology). HEK293T cells ($2 \times 10^6$) were seeded in a 100-mm cell culture dish and transfected with plasmids using PolyJet in vitro DNA transfection reagent PEI (40815ES03, Yeasen Biotechnology). The transfection experiments were performed according to the manufacturer's instructions. For the overexpression of MET proteins, we co-transfected the HEK293T cells with the plasmids of pMD2.G, psPAX2, and pHAGE-Flag-MET to generate lentivirus. For lentivirus infection, 500 µl of virus-containing medium was added to the $5 \times 10^4$ VCaP cells with 0.1% poly-brene. All oligonucleotide sequences used in this study are provided in Supplementary Data 10.

## Bulk RNA-seq library construction and sequencing

Total RNA was extracted from cells using the RNAsimple Total RNA Kit (no. DP419, TIANGEN). Illumina-compatible libraries were prepared using the NEBNext Ultra RNA Library Prep Kit for Illumina (catalog no. E7530L, NEB). In brief, using fragmented mRNA as a template and random oligonucleotides as primers, the first strand of cDNA was synthesized in the M-MuLV reverse transcriptase system. Second-strand cDNA synthesis was subsequently performed using DNA polymerase I and RNase H. The purified double-stranded cDNA undergoes end-repair, A-tailing, and ligation of sequencing adapters. The 250–300-base pair cDNA is screened with AMPure XP beads, PCR amplification is performed, and the PCR products are purified again with AMPure XP beads to obtain the library. After the library is constructed, use the Qubit2.0 Fluorometer for preliminary quantification, dilute the library to 1.5 ng µl$^{-1}$, and then use the Agilent 5400 system to assess the quality of the library. After the insert size meets the expectation, qPCR measures the effective concentration of the library. The qualified libraries were pooled and sequenced on the Illumina platform with a PE150 strategy in Novogene Bioinformatics Technology, according to effective library concentration and data amount required.

## Statistics and reproducibility

All statistical analyses were performed using R (v4.2.3). Data visualization was conducted using ggplot2 (v3.5.1), ComplexHeatmap (v2.14.0), and maftools (v2.22.10). scRNA-seq data were analyzed using Seurat (v4.4.0). Unless otherwise specified, the two-sided Wilcoxon rank-sum test was used to compute the *P*-value, and the Benjamini–Hochberg procedure was applied to compute the adjusted *P*-value. Details of statistical analyses, including sample sizes and test specifications, are provided in the relevant figure legends.

## Reporting summary

Further information on research design is available in the Nature Portfolio Reporting Summary linked to this article.

## Data availability

The genomic and transcriptomic data of CCLE cell lines were obtained from the DepMap data portal (https://depmap.org/portal/)[21]. The somatic mutation and copy number variation data of TCGA and SU2C samples were downloaded from cBioPortal (https://www.cbioportal.org/)[70,71]. The bulk RNA-seq data of TCGA samples were downloaded from the UCSC Xena data portal (https://xena.ucsc.edu/)[72]. The bulk RNA-seq data of metastatic prostate adenocarcinoma patients were from the MET500 cohort[73]. The bulk RNA-seq and scRNA-seq data of PC3 were from GEO (accession numbers GSE116668, GSE157220, and GSE140440)[74–76]. The scRNA-seq data of VCaP and NCI-H660 were downloaded from GEO (accession number: GSE252883)[77]. The scRNA-seq data of normal human prostate were downloaded from GEO (accession number: GSE117403)[69]. For CRPC, the Smart-seq2 scRNA-seq data were downloaded from Single Cell Portal (https://singlecell.broadinstitute.org/single_cell/study/SCP1244), and the others were downloaded from SRA (accession number: PRJNA699369)[78] and GSE210358[47], respectively. The bulk RNA-seq data of the engineered LNCaP cell lines were obtained from GEO (accession numbers: GSE175975 and GSE162225)[50,51]. The bulk RNA-seq data of the engineered VCaP cell lines can be downloaded from SRA (accession number: PRJNA1424369). For prostate cancer organoids, the bulk RNA-seq data were downloaded from GSE199190[57]. The ATAC-seq raw data of metastatic prostate cancer samples, cell lines, and organoids were downloaded from EGA (accession number: EGAS00001006698) and SRA (accession number: PRJNA818767)[52,57]. Source data underlying the figures are provided in Supplementary Data 11. All other data supporting the findings of this study are available from the corresponding author upon reasonable request.

## Code availability

The code is available at GitHub (https://github.com/bioklab/MetPCCellline) and archived at Zenodo (https://doi.org/10.5281/zenodo.18632765)[79].

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

## Acknowledgements

The research is supported by the National Natural Science Foundation of China (Fund 32370715, 82330108), Science Fund for Distinguished Oversea Young Scholars of Shandong Province (2023HWYQ-015), Taishan Young Scholar Program of Shandong Province (tsqn202312020), and Cheeloo Young Scholar Program of Shandong University.

## Author contributions

X.Y.L. and K.L. conceived the study. X.Y.L. performed the majority of computational analysis, Y.G.W. and W.X.Y. performed all the experimental

assays, and K.L. and Y.G.W. supervised the study. X.Y.L., K.L., and X.Y.J. revised the manuscript. All authors contributed to writing, reviewing, and editing the manuscript and approved the manuscript.

## Competing interests

The authors declare no competing interests.
