## [Transparent Peer Review file · Communications Biology]

Multi-omics evaluation of cell lines as models for metastatic prostate cancer

Corresponding Author: Dr Ke Liu

Version 0:

Reviewer comments:

Reviewer #1

(Remarks to the Author)

The manuscript presents a thorough multi-omics assessment of prostate cancer (PC) cell lines as models for metastatic disease. The work addresses an important gap, as metastasis is the lethal stage of PC and guidance on suitable in vitro models remains limited. By integrating genomic and transcriptomic data from bulk tumors, cell lines, and single-cell sources, the study provides valuable biological insights and practical guidance for model selection.

The manuscript is generally well-organized and the findings are supported by the data. Its novelty lies in the focus on metastatic disease and the integrative use of multiple datasets. However, some areas could be improved to strengthen clarity and impact, particularly clearer biological interpretation of results, acknowledgment of model limitations from a biological perspective, and improved readability.

Comments

1. The manuscript presents several analyses, but the biological meaning could be explained more clearly so non-computational readers can follow the significance. This applies to tests such as transcriptomic correlation, scRNA-based subtype matching, MSI/hypermutation calls, mutation hotspot mapping, and immune deconvolution (TIDE/xCell).
2. Some key terms and subtypes need clearer biological context. For example, the "mesenchymal stem-like prostate cancer (MSPC)" subtype should be defined in biological terms (e.g., marker profiles, AR loss, stemness features). Similarly, when noting that PC3 does not cluster with metastatic subtypes, it would help to link this to its known traits, such as AR-negativity and lack of basal markers.
3. The study evaluates a solid set of models, but the coverage remains limited. The patient-derived organoids (PDOs) are few in number (only ten), and conclusions about the MSPC subtype rely heavily on them. Similarly, the engineered LNCaP and VCaP models tested here did not recapitulate MSPC, but these represent only a small subset of possible approaches. It would strengthen the paper if the authors acknowledge these limits and note that additional PDOs, PDXs, or alternative strategies are needed to fully validate models of the MSPC subtype.
4. The link between hypermutation and immune infiltration is well shown in patient data, but in vitro models cannot reproduce immune or stromal interactions. The authors should note that while hypermutated cell lines can model tumor-intrinsic consequences (e.g., genomic instability), they cannot capture immune responses without complementary systems such as co-culture.
5. To make the findings more accessible, the authors could add a brief summary table or schematic that maps which models (cell lines, engineered lines, PDOs) best represent each prostate cancer subtype or stage. This would provide a clear take-home message.
6. The manuscript would benefit from thorough language and grammar polishing.

These refinements will improve readability and presentation. The core content and conclusions remain strong, and addressing the points above will further enhance the clarity and impact of the study.

Reviewer #2

(Remarks to the Author)

This manuscript: "Multi-omics evaluation of cell lines as models for metastatic prostate cancer" presents a comparison between prostate cancer cell lines and metastatic tumor samples using large-scale public datasets (TCGA, SU2C, CCLE, and multiple scRNA-seq cohorts). The authors aim to evaluate the representativeness of prostate cancer cell lines as

models for metastatic disease. They integrate genomic, transcriptomic, and copy-number variation data, revealing key mismatches between cell lines and patient samples. Importantly, they identify VCaP and LNCaP as better models for metastatic prostate adenocarcinoma and hypermutated tumors, respectively, while demonstrating that the widely used PC3 cell line poorly reflects any authentic metastatic subtype. The authors also assess engineered lines and patient-derived organoids.

This study aims at providing guidance for selection of prostate cancer cell lines to model metastatic disease biology. This study could help the prostate cancer research community to better identify high fidelity in vitro models of metastatic disease. However, I find limitations in the novelty and approach. Please see critics and suggestions below.

Major critics

1. Analyses of transcriptional profiles (including regulatory network analyses) comparing prostate cancer cell lines available from CCLE (including LNCaP, VCap and PC3) against metastatic prostate cancer patients (i.e., SU2C) has been already performed and published. See Vasciaveo, et al. Cancer Discovery 2023, PMID: 36374194. Specifically, see analysis in Supplementary Figure S7.

2. The authors claims that other similar studies relied only on transcriptomics data. However, their manuscript does not include other data modalities other than transcriptomics and genomics. There is availability of epigenetic, histone modification, proteomics and metabolomics data through CCLE. The inclusion of some of those data modalities could corroborate their findings. See CCLE or depmap.org for cell line data and dbGap repository for CRPC data from metastasis. Additionally, there is plenty of epigenetics data (e.g., ATAC-Seq data) from metastatic CRPC already published (see Tang, et al. Science 2022, PMID: 35617398) that can be used to compare epigenomics profiles from prostate cancer patient metastasis with epigenomics data from prostate cancer cell lines from CCLE.

Please see also other single-cell data (both transcriptomics and epigenomics, i.e., RNA and ATAC Seq) from some prostate cancer cell lines: PMID: 34489465, Taavitsainen, et al. Nature Communication 2021.

I suggest using these data for a more comprehensive multi-omics analysis.

3. Hypermutation analyses are based on a very small number of cases (e.g., four hypermutated TCGA samples and ten in SU2C), limiting confidence in the TME and CD8⁺ T-cell findings.

4. Major findings about PC3 line were done by looking at the literature (PubMed) which is not an -omics data modality, and by using transcriptomics correlations (Spearman's correlation on gene expression profiles) from CCLE data, which contains n=1 data point per cell line, limiting the reproducibility of the findings derived from quantitative data analysis.

5. The authors discuss tumor and cell line heterogeneity and performed some single cell RNASeq analysis. Given the availability of single-cell data from metastatic prostate cancer patients, it would be interesting to see how single-cell data from these prostate cancer cell lines compare to the distinct cell programs found in metastatic tumors from patients.

Version 1:

Reviewer comments:

Reviewer #1

(Remarks to the Author)

I am happy with the Author's replies as they addressed all my comments. I move to accept the paper. congrats.

Reviewer #2

(Remarks to the Author)

I have reviewed the revised manuscript and the authors' responses. The authors have adequately addressed all of my comments, and I have no further concerns.

We thank the two reviewers for their professional and constructive comments. We have revised our manuscript accordingly and all the revisions are highlighted in yellow. In addition, the revisions made on figures and tables are summarized in the tables at the end of this file. The point-by-point responses are listed as follows:

Reviewer 1

1. The manuscript presents several analyses, but the biological meaning could be explained more clearly so non-computational readers can follow the significance. This applies to tests such as transcriptomic correlation, scRNA-based subtype matching, MSI/hypermutation calls, mutation hotspot mapping, and immune deconvolution (TIDE/xCell).

--Thank you for pointing this out. We have revised our manuscript to include more explanations on the computational methods so that non-computational readers can follow the significance. The revisions are listed as follows:

MSI/hypermutation calls: **see page 7, line 180.**

mutation hotspot mapping: **see page 18, line 490.**

immune deconvolution: **see page 19, line 501.**

transcriptomic correlation: **see page 19, line 509.**

scRNA-based subtype matching: **see page 21 line 549.**

Please note that after further literature review, we realized that we might have used the term “mutation hotspot” improperly in the original version. The term “mutation hotspot” refers to genomic regions that are inherently prone to mutation (see Nestal *et al.*, *Trends in Genetics*, 2021; PMID: 33199048), whereas “hotspot mutation” refers to recurrently observed mutations across patients (see Li *et al.*, *Nature*, 2021; PMID: 32386297). Although these two terms have been used interchangeably (e.g., Miller *et al.*, *Cell Systems*, 2015; PMID: 27135912; Gauthier *et al.*, *Nucleic Acids Research*, 2016; PMID: 26590264), we now adopt a precise nomenclature: “hotspot mutation” is used exclusively for recurrently altered mutations across tumor samples, and “mutation hotspot” is reserved for regions that are intrinsically hypermutable. To maintain scientific rigor and clarity, we have corrected the terminology throughout the revised manuscript and figure legend. Importantly, this correction DOES NOT affect any of the scientific conclusions of our study.

2. Some key terms and subtypes need clearer biological context. For example, the “mesenchymal stem-like prostate cancer (MSPC)” subtype should be defined in biological terms (e.g., marker profiles, AR loss, stemness features). Similarly, when noting that PC3 does not cluster with metastatic subtypes, it would help to link this to its known traits, such as AR-negativity and lack of basal markers.

--Thank you for your comment. We revised our manuscript to include clear biological context at the first mention of MSPC (see **page 10, line 272**). In our original version, we only highlighted the lack of basal markers of PC3. In the revised manuscript, we have added some texts to emphasize its AR-negativity (see **page 11, line 281**).

3. The study evaluates a solid set of models, but the coverage remains limited. The patient-derived organoids (PDOs) are few in number (only ten), and conclusions about the MSPC subtype rely heavily on them. Similarly, the engineered LNCaP and VCaP models tested here did not recapitulate MSPC, but these represent only a small subset of possible approaches. It would strengthen the paper if the authors acknowledge these limits and note that additional PDOs, PDXs, or alternative strategies are needed to fully validate models of the MSPC subtype.

-- We thank the reviewer for highlighting this limitation. We acknowledge that our study does not cover all prostate cancer models, as our primary focus here was on cell lines. As noted, it is important to perform a more comprehensive evaluation of additional models (such as PDXs or models generated using alternative strategies), which we plan to pursue in future studies. We have revised the Discussion section to address this limitation (see **page 17, line 463**).

4. The link between hypermutation and immune infiltration is well shown in patient data, but in vitro models cannot reproduce immune or stromal interactions. The authors should note that while hypermutated cell lines can model tumor-intrinsic consequences (e.g., genomic instability), they cannot capture immune responses without complementary systems such as co-culture.

-- We thank the reviewer for this comment. Indeed, cell lines alone cannot fully capture immune responses, which is a common limitation of this model type. We focused on tumor-intrinsic characteristics because they play a pivotal role in shaping immune responses (see Abida *et al.*, *JAMA Oncology* 2019, PMID:30589920). When placed in a microenvironmental context (e.g., co-culture or xenograft models), hypermutated cell lines should better model hypermutated prostate cancer than non-hypermutated ones. We have revised the Discussion section to highlight this point (see **page 15, line 413**).

5. To make the findings more accessible, the authors could add a brief summary table or schematic that maps which models (cell lines, engineered lines, PDOs) best represent each prostate cancer subtype or stage. This would provide a clear take-home message.

--Thank you for your suggestion. We added a summary figure to summarize the mappings and this does provide a clear take-home message (see **Supplementary Fig. 8**).

6. The manuscript would benefit from thorough language and grammar polishing.

--Thank you for your suggestion. We have asked professional English speakers to help us revise the manuscript. In addition, we also utilized AI tools to polish the language and grammar.

Reviewer 2

1. Analyses of transcriptional profiles (including regulatory network analyses) comparing prostate cancer cell lines available from CCLE (including LNCaP, VCap and PC3) against metastatic prostate cancer patients (i.e., SU2C) has been already performed and published. See Vasciaveo, et al. Cancer Discovery 2023, PMID: 36374194. Specifically, see analysis in Supplementary Figure S7.

--Thank you for directing us to this literature. We carefully reviewed the study by Vasciaveo *et al.* (especially the analysis in Supplementary Fig. S7) and agree that OncoMatch provides valuable insight into model selection for prostate cancer patients. The OncoMatch method is based on bulk RNA-seq profiles and aims to identify “cognate” cell line models matched to patient tumors. However, our work is not a duplication of their analysis.

a. Our study emphasizes the importance of prostate cancer heterogeneity in selecting appropriate models. For example, we show that hypermutated prostate cancers exhibit higher CD8⁺ T-cell infiltration and that LNCaP is the most suitable cell line for modeling this context. For NEPC, we identify NCI-H660 as the most appropriate model. Such subtype-specific guidance is not provided in Vasciaveo *et al.*'s study.

b. Our integrated evaluation across genomic, transcriptomic, and epigenomic layers uncovers new findings. We demonstrate the limitations of PC3 in recapitulating MSPC and provide both RNA-seq and ATAC-seq evidence. For ARPC, we show that VCaP is a superior model because it displays high similarity to patient tumors in our multi-omics correlation analyses and meanwhile carries strong AR amplification—a hallmark of metastatic prostate cancer.

c. Our study further incorporates patient derived organoids (PDOs) and highlights their unique value in modeling MSPC, which was not addressed in Vasciaveo *et al.*'s study.

In summary, our work provides a more comprehensive and in-depth framework for selecting prostate cancer models, extending beyond what was covered by Vasciaveo *et al.* We appreciate their contribution and have now cited their study in the revised manuscript (see page 4, line 101).

2. The authors claims that other similar studies relied only on transcriptomics data. However, their manuscript does not include other data modalities other than

transcriptomics and genomics. There is availability of epigenetic, histone modification, proteomics and metabolomics data through CCLE. The inclusion of some of those data modalities could corroborate their findings. See CCLE or depmap.org for cell line data and dbGap repository for CRPC data from metastasis. Additionally, there is plenty of epigenetics data (e.g., ATAC-seq data) from metastatic CRPC already published (see Tang, *et al.* Science 2022, PMID: 35617398) that can be used to compare epigenomics profiles from prostate cancer patient metastasis with epigenomics data from prostate cancer cell lines from CCLE. Please see also other single-cell data (both transcriptomics and epigenomics, i.e., RNA and ATAC Seq) from some prostate cancer cell lines: PMID: 34489465, Taavitsainen, *et al.* Nature Communication 2021. I suggest using these data for a more comprehensive multi-omics analysis.

-- Thank you for this constructive suggestion. Actually, the Tang *et al.* 's data has already been used in the original version of our manuscript (the bulk RNA-seq data of organoids comes from this study); in the revisited version, we additionally incorporated the ATAC-seq data of organoids and cell lines presented by them. Because the prostate cancer cell lines mentioned by Taavitsainen *et al.* (LNCaP and VCaP) are already covered by the Tang *et al.* 's dataset, we did not further include their data in the revision. In addition, after an extensive search across public domain, we identified a dataset containing both bulk RNA-seq and ATAC-seq data of metastatic prostate cancer patient samples (Raunak, *et al.*, *Cancer Research* 2024, PMID: 38990734). We summarize our analysis based on these data as follows:

a. We performed PCA on ATAC-seq data of prostate cancer cell lines and observed three major clusters, which were highly concordant with RNA-seq-based PCA analysis. In addition, enrichment analysis of the top 1,000 peaks with highest PC1-loading showed significant enrichment of EMT, *IL-6/JAK/STAT3*, and *IL-2/STAT5* signaling pathways, which is also consistent with our RNA-seq findings (see page 12, line 321 and Fig. 5a-b).

b. We computed chromatin-accessibility similarity between cell lines and patient samples, and the results highly mirrored our findings derived from transcriptomic analysis: ARPC samples showed the highest similarity to VCaP while NEPC samples showed the highest similarity to NCI-H660. As expected, MSPC samples exhibited moderate similarity to PC3 and this is consistent with our conclusion that PC3 only partially recapitulates MSPC biology (see page 12, line 328 and Fig. 5c-e).

c. We computed chromatin-accessibility similarity between patient-derived organoids (PDOs) and MSPC patient samples and further confirmed our conclusion that PDOs more faithfully recapitulate the MSPC than PC3. Especially, MSKPCa12, which has the highest transcriptomic similarity with patient samples, also has the highest

chromatin-accessibility similarity (see page 14, line 374 and Supplementary Fig. 6h).

In summary, our findings based on ATAC-seq data are highly consistent with that derived from RNA-seq data analysis. We thank the reviewer again for encouraging a deeper multi-omics analysis, which has significantly strengthened the rigor of our conclusions.

3. Hypermutation analyses are based on a very small number of cases (e.g., four hypermutated TCGA samples and ten in SU2C), limiting confidence in the TME and CD8⁺ T-cell findings.

--Thank you for raising this important point.

a. We agree that the number of hypermutated cases is limited, and this appears to be an inherent characteristic of prostate cancer (see Abida *et al.*, *JAMA Oncology* 2019, PMID:30589920). However, it is worth noticing that we have 490 non-hypermutated cases, suggesting the estimation of the null distribution should be accurate, which can help us effectively control the type I error in our hypothesis testing.

b. Although DESeq2 is one of the most popular methods for differential gene expression analysis, it internally utilized negative binomial distribution to model the data. Therefore, in our original version we further compared the expression of *CD3D*, *CD3E*, *CD8A*, *GZMA*, *GZMB*, *PRF1*, *PDCD1* and *CTLA4* using Wilcoxon rank-sum test (see page 8, line 203 and Supplementary Fig. 2c). This test is a conservative, non-parametric statistical method and DOES NOT require any assumptions on data distribution.

Based on the above two points, we believe our findings about the CD8⁺ T cell infiltration are confident.

4. Major findings about PC3 line were done by looking at the literature (PubMed) which is not an -omics data modality, and by using transcriptomics correlations (Spearman's correlation on gene expression profiles) from CCLE data, which contains n=1 data point per cell line, limiting the reproducibility of the findings derived from quantitative data analysis.

--Thank you for this important comment. To address the reviewer's concern about reproducibility and reliance on n=1 CCLE samples, we performed additional analyses that independently confirm our conclusions.

a. We re-examined the expression of *AR* (ARPC marker), *SYP* (NEPC marker), and *CD44* (MSPC marker) in PC3 using bulk RNA-seq data and scRNA-seq data from

independent non-CCLE datasets, and confirmed that PC3 was a line of MSPC subtype (see page 11, line 283 and Supplementary Fig. 4a-e).

b. When evaluating transcriptomic similarity between MSPC malignant cells and CCLE cell lines, we incorporated the external bulk RNA-seq PC3 samples into the analysis and the results remained consistent: PC3 was still NOT the cell line with the highest ranking. We further validated this observation using an independent scRNA-seq dataset from metastatic prostate cancer (see page 12, line 309 and Supplementary Fig. 4m and Supplementary Fig. 5a-d).

c. In our original manuscript we showed that PC3 expressed very low levels of basal markers and this feature may explain the low similarity between PC3 and MSPC tumors. To strengthen this conclusion, we re-examined the expression of basal markers in an independent PC3 scRNA-seq dataset and derived similar results (page 11, line 307 and Supplementary Fig. 4l).

5. The authors discuss tumor and cell line heterogeneity and performed some single cell RNAseq analysis. Given the availability of single-cell data from metastatic prostate cancer patients, it would be interesting to see how single-cell data from these prostate cancer cell lines compare to the distinct cell programs found in metastatic tumors from patients.

--Thank you for this insightful suggestion. We performed differential gene expression analysis between cell line and patient samples based on scRNA-seq data, and then performed enrichment analysis utilizing the 50 cell programs of MSigDB Hallmark (see page 17, line 443 and Supplementary Fig. 7a-f).

We found that cell cycle related programs (such as “Mitotic Spindle”, “G2-M Checkpoint”, “E2F targets”) were consistently up-regulated in cell line (regardless of subtype), suggesting cell lines have more active cell cycle.

The “TNF-alpha Signaling via NF-kB” was down-regulated in both PC3-vs-MSPC and NCIH660-vs-NEPC comparisons, which may be not surprising since cell lines alone cannot effectively model the immune microenvironment. Especially, the “Interferon Gamma Response” and “Interferon Alpha Response” were specifically down-regulated in the PC3-vs-MSPC comparison. Another interesting observation is that the “Hypoxia” program is down-regulated in PC3-vs-MSPC and NCIH660-vs-NEPC comparisons, which reflect the fact that malignant cells were in an environment lack of oxygen. Consistent with this, we also detected the “Oxidative Phosphorylation” program was down-regulated in the VCap-vs-ARPC and NCIH660-vs-NEPC comparisons.

In summary, we found the transcriptomic divergence mainly reflects the differences of

the microenvironment and this has been to be a common limitation of cell line. Actually, reviewer 1 also suggested us to mention the limitation of cell line and we have revised the discussion accordingly (see page 15, line 413).

Main Figure:

Figure 1	unchanged
Figure 2	unchanged
Figure 3	unchanged
Figure 4	unchanged
Figure 5	added in the revision
Figure 6	In our previous manuscript, this figure was Figure 5 , now Figure 6 in the revised version.

Supplementary Figure:

Supplementary Figure 1	unchanged
Supplementary Figure 2	a: added in the revision b: previous panel a c: previous panel b d: previous panel c
Supplementary Figure 3	unchanged
Supplementary Figure 4	a-c: added in the revision d: previous panel a e: added in the revision f-k: previous panel b-g l-m: added in the revision
Supplementary Figure 5	added in the revision
Supplementary Figure 6	previous Supplementary Fig 5 a-g: unchanged h: added in the revision
Supplementary Figure 7	added in the revision
Supplementary Figure 8	added in the revision

Supplementary Table:

Supplementary Table 1-8	unchanged
Supplementary Table 9	added in the revision